# Testing for a Random Walk Structure in the Frequency Evolution of a Tone in Noise

**DOI:** 10.3390/s22166103

**Published:** 2022-08-15

**Authors:** Scarlett Abramson, William Moran, Robin Evans, Andrew Melatos

**Affiliations:** 1Department of Electrical and Electronic Engineering, University of Melbourne, Parkville, VIC 3010, Australia; 2School of Physics, University of Melbourne, Parkville, VIC 3010, Australia

**Keywords:** model verification, structure identification, random walk frequency, sinusoidal signal

## Abstract

Inference and hypothesis testing are typically constructed on the basis that a specific model holds for the data. To determine the veracity of conclusions drawn from such data analyses, one must be able to identify the presence of the assumed structure within the data. In this paper, a model verification test is developed for the presence of a random walk-like structure in the variations in the frequency of complex-valued sinusoidal signals measured in additive Gaussian noise. This test evaluates the joint inference of the random walk hypothesis tests found in economics literature that seek random walk behaviours in time series data, with an additional test to account for how the random walk behaves in frequency space.

## 1. Introduction

Many analyses done today are model-based and inference results drawn from these works rely heavily on strong assumptions to hold. One may ask if the models used are the correct choice; and if true, or if not, what impact that would have on the veracity of the results. The questions surrounding verifying decisions in model selection have been popular fields of inquiry. (See [1,2] for thorough reviews of model selection methods.)

This paper considers a specific structure identification (SID) problem: to determine if a given time series arose from noisy samples of a sinusoid whose frequency varies according to a random walk (RW). This problem has important practical applications in several areas of signal processing as discussed briefly in Section 1.1 and is part of the long-established theoretical tradition of optimal detection and estimation for tones with randomly varying frequency. Our current motivation relates to the, as yet unsuccessful, search for continuous gravitational waves (CGWs). The structure of the frequency variation of such signals has astrophysical significance.

Since the seminal works of Kolmogorov and Wiener in the 1940s, the majority of sensor-type signal processing approaches are model-based. Sensor-type signal processing techniques based on carefully defined models of sensors and signals have two huge advantages—firstly, optimal processing methods can be developed by exploiting model structure, and secondly, performance can often be characterized analytically.

There are, however, certain disadvantages. If the model is not an accurate description of the sensor, the signals, or both, then the above advantages can be diminished.

To address this issue, over the past 50 years many techniques for identifying parameterized models of given structure from measured data have been developed including adaptive methods which continually adjust the model parameters in real-time based on the most recent data. Robust techniques have been developed to maintain near-optimal performance under fairly broad modelling errors including classes of structural errors.

The next important step in model-based sensor and signal processing problems is the validation of models. In some situations, model validation can be based on ground truth, though this is rarely possible.

Recently, there has been growing interest and activity in quantifying the evidence for a model which is being used in a model-based optimal-sensor signal processing scenario. That is, given a model and an optimal processing solution—there is interest in determining the evidence for the model being used based on observed data and optimally-processed outputs. A comprehensive review covering methods for evaluating evidence for a model can be found in the recent paper [3].

The theoretical foundations of general SID (as opposed to parameter estimation) are still not well developed. Existing approaches such as Bayes factor-based methods, model residual testing, and multiple hypotheses testing all propose a structure, and then test in different ways how strongly the data are consistent with the model. While this is an advance on parametric estimation for a fixed given model, it is still a long way from true structural discovery.

For the problem considered in this paper, an initial question is how to even determine whether an RW is present in the data. This is a well-researched field in the econometrics literature known as random walk hypothesis (RWH) testing [4]. Such methods try to determine from a recording of time-series data whether the data behaves as an RW.

The second question is then how to distinguish between an RW and the trigonometric function of an RW. One key quality that comes into play here is that the resulting process is bounded so it is trivial to do this given sufficiently a long recording time. However, many processes are random and bounded and the RWH techniques would not distinguish the raw trigonometrically transformed data by design. To distinguish, one must pick out the component believed to act as an RW (the frequencies of the signal). Conventionally this is done by the Fourier transform.

Once in Fourier space, one may then naively think to simply apply RWH testing techniques and yield a viable result. A challenge comes from the fact that frequency space is cyclic. Given that, it then becomes a challenge to distinguish between something that would resemble an RW under RWH testing and the Fourier transform of white Gaussian noise (WGN).

We argue here that should the trigonometrically transformed-RW in question vary slowly, it could be distinguished from the Fourier transform of WGN and so be susceptible to RWH testing methodologies.

We will present a non-nested multiple hypothesis procedure to test if a structure (RW frequency signal) is a good model. The choice of hypotheses is important in order to avoid structures that are RW-like but not actually RWs. We create four hypotheses which enable sharp discrimination between RW structure and structures which generate data close to an RW in frequency space.

For completeness, we briefly outline a number of approaches to SID. We then explain the logic underlying the multiple hypothesis structure we analyse in this paper.

Bellman & Åström [5] set out a criterion for structural identifiability, and a generalised approach to SID problems. However, a key challenge is the arbitrary nature of setting criteria for accepting a given model over others while minimising complexity. One of the earliest attempts to deal with this complex problem is the Akaike Information Criterion (AIC) proposed in [6], based on a form of dimension-penalization. However, as Rissanen argued [7], Akaike’s criterion only minimizes total predictive error in the degenerate case where one model size yields an estimate with probability one. Rissanen proposed the Minimum Descriptor Length (MDL) criterion [7], which is invariant under linear coordinate transformations and consistent with respect to dimension estimation. Additionally, Willems formalized SID problems through a behavioural approach [8] that asks ‘given inputs and outputs to a system, does any structure exist at all which may model such behaviours?’ ‘Tearing’ [framing the system as a network of interconnected subsystems], ‘zooming’ [modelling the subsystems], and ‘linking’ [modelling the connections between subsystems] offer systematic approaches to the task. More recently, kernel-based methods implemented in the machine learning methodology have been a popular tool (see [9] for a thorough review). Few of them have addressed the issue of generating hypothesis tests on the model structure when applied to an observed phenomenon. Kernel methods test whether a structure of some sort works, without identifying it explicitly [9]; this is not what we seek to do. Rather, we ask whether a specific structure accurately models an observed phenomenon in a statistically significant sense.

We explore an SID problem, which we phrase through the lens of model detection. This is intrinsically different from a signal detection problem with a known (or more appropriately assumed) signal model. We wish to decide whether any sinusoidal signal with random time-based variations in its instantaneous frequency can provide a fit, with confidence, to the given data. This, rather, seeks the presence of a structure in a record, not the presence of a signal with a specified structure. However, what we observe is the signal itself, and not its underlying parameters. Furthermore, here we seek whether the data fits into a class of models, and so do not impose any specific model (target signal) within the said class. Thus, given that within the class the model parameters may be arbitrary, we assume that they are unknown. In the case of detection, with unknown parameters, we then must implement estimation procedures to verify whether the data resembles a sinusoidal signal with time variation in its tone.

The parameter estimation problem of optimally estimating the frequency of a constant frequency tone based on noisy measurements of the tone is among the well-studied problem in signal processing with many papers exploring its finer details. The general problem itself was formulated in discrete time by Rife and Boorstyn [10], following the work of Slepian [11] in continuous time. In approaching our problem, we assume that the instantaneous frequency varies linearly over short time intervals (snapshots, blocks) and follows an RW structure from snapshot to snapshot. A potential extension would be block-wise (in time) frequency modulation of arbitrary polynomial order (with respect to time); this would, in effect, extend the problem to any continuous frequency modulation.

The statistical theory surrounding RWH testing [12] has been shown to be effective at deciding in certain problems whether the stock market can be said not to be following an RW [13,14,15], with implications for forecasting of market trends.

However, it is argued that such tests can lead to ambiguous conclusions [16] (in the sense of what conclusions drawn mean with regards to the question) or inconsistent results [17] (in the sense of robustness). This is thought to arise from the sensitivity of variance-based inference under the small-sample regimes [18]. Furthermore, because of phase-wrapping behaviours, as discussed in the proof of Remark 1, such methods do not discriminate between a signal measured in noise with an RW in frequency and WGN.

The problem that we consider involves the detection of a model of a trigonometric transformation of an RW process, rather than the detection of an RW process. This is inherently different to standard RWH testing, as trigonometric transformations are nonlinear in nature. The plausibility of testing the RWH under such a transformation has been considered by other researchers in the economics literature [19,20].

An auto-regressive (AR) time series ${\left\{\xi_{n}\right\}}_{n}$ of order 1, known as an AR(1) process, has a recursive representation of the form $\xi_{n}=\rho \xi_{n-1}+\varsigma_{n},$ with independent and identically distributed (IID) driving noise ${\left\{\varsigma_{n}\right\}}_{n}$. The Dickey-Fuller (DF) unit root test [15], assuming the presence of an AR(1) process ${\left\{\xi_{n}\right\}}_{n}$ (not corrupted by noise) in the data, tests a null hypothesis of a stationary (in the statistical sense) first-difference process ($\xi_{n}-\xi_{n-1}$) which would occur if |ρ|=1 against an alternative hypothesis |ρ|≠1.

It had been shown [20] that the DF test, if applied to the trigonometric transformation of an AR(1) process—asymptotically may still be used to test for stationarity of the first difference process—following analytical [21], and empirical [19] arguments on the asymptotics of nonlinear transformations on 1st-order integrated time series I(1), and subsequently on AR(1) processes.

However, this is different to the model we test for: the signal model of a scaled trigonometric transformation of an AR(1) process with |ρ|=1, given noise-corrupted measurements. As we do not assume, inherently, that a (transformed) AR(1) time series is present within the data, but rather are testing the validity of such a model—given the data—the DF test regime does not apply here, and so we follow a different scheme that will be discussed below.

In this paper, we advance a scheme for discriminating between WGN and a sinusoidal signal whose instantaneous frequency executes a discrete-time RW, which, when combined with RWH methods provides a more general inference on the existence of an RW structure in frequency. In Section 2.1 the problem is formulated mathematically. We design an estimator-detector in Section 2.2 and Section 2.3, and discuss its analytical performance in Section 3.1 and Section 4.

In continuous time the RW is replaced by a Wiener process and in this context, prior to sampling, we are faced with signals of infinite bandwidth. This is, as is often the case with mathematical models, a useful idealization. Whereas a typical function of a Wiener process has infinite bandwidth, ultimately the signal has to be measured and this forces a bandwidth constraint on the measured signal imposed by the sensor. This has to be taken into account in the sampling and subsequent processing of the discretized signal. We work within the discrete time domain here and the bandwidth issues are overcome by the determination of the sampling rate of the continuous time signal.

### 1.1. Applications

The concept of a time-varying auto-regressive (TVAR) tone appears across various fields. Below are some applications of TVAR tones in the literature, including the motivating application for this paper in gravitational wave (GW) astronomy.

#### 1.1.1. Gravitational Wave Astronomy

Predicted theoretically by Einstein in 1915, GWs are disturbances in space and time, generated by the acceleration of asymmetric bodies, which propagate at the speed of light [22] as perturbations of the metric tensor $g_{\mu \nu }$, which describes distances in spacetime by the invariant interval $ds^{2}=g_{\mu \nu }dx^{\mu }dx^{\nu }$ (with respect to displacement four-vectors $dx^{\mu }$).

GWs cause oscillations in the proper displacements of freely falling test masses and can be detected by long-baseline laser interferometers, such as the Laser Interferometer Gravitational-Wave Observatory (LIGO) [23].

A GW incident normally on the plane of an interferometer stretches and squeezes distances along the arms [22]; along two characteristic polarisations ‘plus’ (whose principal axes of action align with those of the spatial dimensions) and ‘cross’ (whose principal axes of action are rotated $\frac{\pi }{4}$ with respect to those of the spatial dimensions).

At the time of writing, dozens of GW signals have been detected, including the first discovery of the chirped tones from a binary black hole merger in 2015 [24] and a binary neutron star merger in 2017 [25], inaugurating a new era in astronomy.

Instruments such as LIGO search for a variety of waveforms, not just chirps from mergers. The theory also predicts the existence of persistent, sinusoidal, quasimonochromatic CGW signals, believed to originate (for example) from isolated or binary neutron stars [26]. Under the biaxial rotor model (a rigid body with two equal principal moments of inertia), a pulsar emits GWs continuously at $f_{☆}$ and 2$f_{☆}$ [27].

For an isolated neutron star, $f_{☆}$ decays monotonically over timescales $≳10^{3}$ years, but wanders stochastically on smaller timescales [28]; suspected mechanisms include seismic activity in the crust or far-from-equilibrium avalanche processes in the superfluid interior.

Whereas, in pulsar binaries, the stochastic wander of spin frequency is driven by fluctuations in the accretion torque of gas from the binary companion [29,30]; the underlying process is suspected to be a result of transient disk formation [31], or instabilities in the disk-magnetosphere [30].

The modelling and detection of such CGWs by interferometric data (such as from aLIGO) follows the work of Jaranowski [27] (and the following parts). The stochastic wander in $f_{☆}$ is incorporated into detection schemes based on hidden Markov models [32,33].

#### 1.1.2. Structural Analysis

Materials used in construction exist in meta-stable states, and their properties evolve stochastically over long timescales because of a variety of factors (such as cracking or the influence of humidity). To ensure structural stability one must be aware of the material’s natural frequency and harmonics, to damp against excitation such as from applied wind pressure (for example, the case of the Tacoma Narrows Bridge, “Galloping Gertie”) or from synchronous loading (as seen with the London Millennium Bridge). When considering concrete elements, for example, such a natural frequency wanders randomly in time. Efforts are made to track this randomly wandering natural frequency in papers such as [34].

#### 1.1.3. Sound Processing

The tracking of TVAR harmonics in signals is an approach considered in communications and sound processing problems such as in the tracking of a time-varying chirp (such as the Doppler shifts in mobile communications). Such applications have been discussed in papers such as [35] and others.

## 2. Materials and Methods

### 2.1. Problem Formulation

To begin, we denote the discretisation (for indices n=0,…,N−1, k=0,…,K−1) of a general temporal function [·] by:(1)$$\begin{array}{cc} {\left[\cdot \right]}_{n}^{\left(k\right)} & =\left[\cdot \right]\left(t_{n}^{\left(k\right)}\right), \end{array}$$(2)$$\begin{array}{cc} t_{n}^{\left(k\right)} & =[n+k(N-1)]T, \end{array}$$
so that $t_{N-1}^{\left(k\right)}=t_{0}^{\left(k+1\right)}.$

This discretisation provides a partition for a data record of data sampled uniformly at rate $f_{s}=\frac{1}{T}$ into *K* blocks ${\left\{x_{n}^{\left(k\right)}\right\}}_{n=0,\ldots ,N-1}^{k=0,\ldots ,K-1}$ of *N* observations, that is,
$${\left\{x_{n}^{\left(k\right)}\right\}}_{n}^{k}=\left\{{\left\{x_{n}^{\left(0\right)}\right\}}_{n},\ldots ,{\left\{x_{n}^{\left(K-1\right)}\right\}}_{n}\right\}=\left\{\left\{x_{0}^{\left(0\right)},\ldots ,x_{N-1}^{\left(0\right)}\right\},\ldots ,\left\{x_{0}^{\left(K-1\right)},\ldots ,x_{N-1}^{\left(K-1\right)}\right\}\right\}$$
(where in the temporal discretisation structure noted above, we overlap the last observation $x_{N-1}^{\left(k^{\prime }\right)}$ and first observation $x_{0}^{\left(k^{\prime }+1\right)}$ of any two adjoining blocks $k=k^{\prime },k^{\prime }+1$) to a data record of K(N−1)+1 observations, or KN points including repeated points.

For a block ${\left\{x_{n}^{\left(k\right)}\right\}}_{n}$ (fixed *k*), we consider *N* to be the sample size of that subset
$$\{x_{n^{\prime }=\left(k-1\right)\left(N-1\right)+0},x_{n^{\prime }=\left(k-1\right)\left(N-1\right)+1},\ldots ,x_{n^{\prime }=\left(k-1\right)\left(N-1\right)+\left(N-1\right)}\},$$
of the full record
$$\{x_{n^{\prime }=0},x_{n^{\prime }=1},\ldots ,x_{n^{\prime }=K\left(N-1\right)}\}.$$

The parameter *N* is the *blockwise sample size*. Graphically, the discretisation (with overlap at endpoints) is constructed as in Figure 1.

Now, for any block (fixed *k*), let ${\left\{w_{n}^{\left(k\right)}\right\}}_{n}$ be samples of complex WGN ($V\left[{\left\{w_{n}^{\left(k\right)}\right\}}_{n}\right]=\sigma_{w}^{2}<\infty$ known), and let ${\left\{s_{n}^{\left(k\right)}\right\}}_{n}$: $s\left(t_{n}^{\left(k\right)}\right)=s_{n}^{\left(k\right)}$ be a sampled complex sinusoid defined by,
(3)$$\begin{array}{cc} s_{n}^{\left(k\right)} & =Ae^{i\phi_{n}^{\left(k\right)}}, \end{array}$$
where, for fixed *k*, ${\left\{\phi_{n}^{\left(k\right)}\right\}}_{n}$: $\phi \left(t_{n}^{\left(k\right)}\right)=\phi_{n}^{\left(k\right)}$ is the instantaneous phase of the sampled sinusoid (assuming a quadratic structure in time). That is, for fixed *k*, ${\left\{f_{n}^{\left(k\right)}\right\}}_{n}$: $f\left(t_{n}^{\left(k\right)}\right)=f_{n}^{\left(k\right)}$ is linearly ramping between $f_{0}^{\left(k\right)}$ and $f_{0}^{\left(k+1\right)}$ with respect to pivoting frequencies ${\left\{f_{0}^{\left(k\right)}\right\}}_{k}$:(4)$$\begin{array}{c} f_{n}^{\left(k\right)}=f_{0}^{\left(k\right)}+\frac{n}{\left(N-1\right)}\left[f_{0}^{\left(k+1\right)}-f_{0}^{\left(k\right)}\right], \end{array}$$
which corresponds to a phase:(5)$$\begin{array}{cc} \phi_{n}^{\left(k\right)}= & \varphi_{{}_{0}}+\frac{1}{2}\frac{n^{2}}{\left(N-1\right)}T\left[f_{0}^{\left(k+1\right)}-f_{0}^{\left(k\right)}\right]\\ \; & +\frac{1}{2}\left(N-1\right)T\left[f_{0}^{\left(k\right)}-f_{0}^{\left(0\right)}\right], \end{array}$$
where $\varphi_{{}_{0}}:=\phi_{0}^{\left(0\right)}$.

**Definition** **1.***We will say that the pivot frequencies ${\left\{f_{0}^{\left(k\right)}\right\}}_{k}$ are* stable *if they vary ‘slowly’ in the sense that,*
(6)$$\begin{array}{cc} |m_{0}^{\left(k+1\right)}-m_{0}^{\left(k\right)}| & \ll N, \end{array}$$
*where, as defined in (Equation 8), ${\left\{m_{0}^{\left(k\right)}\right\}}_{k}$ are the centre frequencies of the Fourier bins corresponding to ${\left\{f_{0}^{\left(k\right)}\right\}}_{k}$ with $-\frac{f_{s}}{2}$≤$f_{0}^{\left(k\right)}$$\leq \frac{f_{s}}{2}.$*

We note that this is effectively a band-limiting statement on the signal. Centre frequencies of Fourier bins of the signal are limited in their variation between consecutive bins to be significantly less than the length *N* of a block. Effectively this prevents wrapping of the frequencies within a block and aliasing.

**Remark** **1.**
*By means of RWH testing, one cannot distinguish between an RW in frequency space that violates condition (Equation 6) and the sequence of peak frequencies ${\left\{f_{w}^{\left(k\right)}\right\}}_{k}$ of blocks (fixed k) of WGN ${\left\{w_{n}^{\left(k\right)}\right\}}_{n}$ when processing via Fourier transforms.*


Existing RWH tests, such as the LOMAC Test [13]; or the Chow-Denning Test [14], are structured around variance ratio (VR) testing (for an overview of such tests see [12]). If a process ${\left\{\varsigma_{n}\right\}}_{n}$ does not satisfy condition (Equation 6), then for any realistic *N*, the existing RWH tests would view the process ${\left\{{\left[f{\left(\varsigma \right)}_{n}\right]}_{f_{s}}\right\}}_{n}$ as it would view the peak frequencies ${\left\{{\left[f{\left(\omega \right)}_{n}\right]}_{f_{s}}\right\}}_{n}$ of blocks of WGN ${\left\{\omega_{n}\right\}}_{n}$.

To clarify, a variance ratio test would not be able to determine whether ${\left\{{\left[f{\left(\varsigma \right)}_{n}\right]}_{f_{s}}\right\}}_{n}$ was drawn from $N(0,{\left[\sigma_{\omega }^{2}\right]}_{f_{s}}),$ or from $N({\left[\mu_{\xi }\left(t\right)\right]}_{f_{s}},{\left[\sigma_{\xi }^{2}\left(t\right)\right]}_{f_{s}})$ (where {ξ(t)} is some RW on *F*).

Another issue that is faced, is the wrapping behaviour of frequency spaces at the edges. If the frequency variations do not occur sufficiently far from the boundaries—then there would also be issues distinguishing an RW frequency process, that is $f_{s}$-modular, from the peak frequencies of a WGN process, that are $f_{s}$-modular.

To clarify, should a process ${\left\{\xi_{n}\right\}}_{n}$ in the frequency space *F* step outside the domain by some small amount ϵ>0: $\epsilon \ll f_{s}$, it would appear to jump to the other boundary. Because of wrapping issues, an RWH testing algorithm, utilising Fourier transforms, would read this as having jumped a length $f_{s}-\epsilon$ either by $\frac{f_{s}}{2}↓-\frac{f_{s}}{2}+\epsilon$ or $-\frac{f_{s}}{2}↑\frac{f_{s}}{2}-\epsilon$, rather than the potentially infinitesimal jump of length ϵ of $\frac{f_{s}}{2}↑\frac{f_{s}}{2}+\epsilon$ or $-\frac{f_{s}}{2}↓-\frac{f_{s}}{2}-\epsilon$. Such a process would then appear to have variance comparable to the size of the domain itself and would be indistinguishable from the frequencies of a WGN process.

**Remark** **2.**
*Recalling that on an unbounded interval, an RW process is also unbounded, and thus the sinusoid with an RW frequency would have infinite bandwidth in the asymptotic case. However, the mere act of measurement is a finite process in finite time—which then imposes a finite bandwidth upon the process.*

*By standard practice, the band-limiting of the signal would be determined by an anti-aliasing filter setting the lower bound on the sampling frequency to the Nyquist limit. Then, one would be able to choose the block-wise sample size determined by the sampling frequency as per condition (Equation 6), and the other parameters can be determined to optimise the efficiency of the verification process.*

*For a more detailed discussion of how to optimally set out the parameters, refer to Section 4.*


We discretise the frequency range $\left[-\frac{f_{s}}{2},\frac{f_{s}}{2}\right]$ into Fourier bins (indexed by m=0,…,N−1) of width $L=\frac{1}{NT}$, and represent the effective frequency f(m) of a bin *m* by:(7)$$\begin{array}{c} f\left(m\right)=\left\{\begin{array}{c} \frac{m}{NT},\;\;m=0,\ldots ,\left\lfloor \frac{N}{2}\right\rfloor -1,\\ \frac{m-N}{NT},\;m=\left\lfloor \frac{N}{2}\right\rfloor ,\ldots ,N-1. \end{array}\right. \end{array}$$

Then the discretisation in time (and subsequently frequency) is equivalent to,
(8)$$\begin{array}{cc} f_{n}^{\left(k\right)} & =f\left(m_{n}^{\left(k\right)}\right)+\delta_{n}^{\left(k\right)}, \end{array}$$
with n=0,…,N−1, and where $\delta_{n}^{\left(k\right)}$ is a displacement term representing uncertainty in phase due to resolution of frequency; if nothing is known about its distribution we assume $\delta_{n}^{\left(k\right)}∼U\left(-\frac{L}{2},\frac{L}{2}\right)$. This is a non-physical prior which will impose little to no bias on inferences, used similarly in discretised methods such as in HMMs [32,33].

**Definition** **2.***We will say that a process ${\left\{\xi_{n}\right\}}_{n}$ is* RW-like *if, when testing it against the RWH, its behaviour would not be considered significantly different from that of an RW, in the statistical sense.**In terms of AR(1) models, of the form $\xi_{n}=\rho \xi_{n-1}+\varsigma_{n},$${\left\{\varsigma_{n}\right\}}_{n}$ IID, we would say that a process ${\left\{\xi_{n}\right\}}_{n}$ is* RW-like *if, when testing it against the RWH, one cannot say in a statistically significant sense that |ρ|≠1.*

The importance of appropriately primed SID approaches on data series assumed to be containing RWs prior to significance tests based on that assumption is made clear by Durlauf and Phillips [36]; who analytically characterised the behaviour of regression coefficients for time trends in data, assuming trend stationary data, on I(1) processes (a class under which RW-like series fall).

They found, analytically, that regressions of an RW against a time trend will yield incorrect inferences of a greater significance of a trend than the present work, resulting in a stronger bias for hypotheses assuming time trends than appropriate; supporting previous empirical works done via Monte Carlo (MC) simulations [37].

For time-series data tested against the RWH, alternative hypotheses of RW-like series may be distinguished from an RW by spectral methods [38] by utilising the weak convergence (in the Hilbert sense, that is $\langle x_{n},y\rangle \to \langle x,y\rangle \forall y$) under the sup metric $d\left(f,g\right)=\sup_{x}\left\{d^{\prime }\left(f\left(x\right),g\left(x\right)\right)\right\}$ of the periodogram deviation process (the sequence of deviations of the time-series in question from the periodogram of a true RW process) to a Brownian bridge on C[0,1].

This is not a convergence that is preserved under trigonometric transforms, and thus we would not be able to apply equivalent tests in our case.

This, taken with Remark 1, yields that the following hypothesis testing structure is enough to detect an RW in frequency space, as per our model:(9)$$\begin{array}{c} \left\{\begin{array}{c} H_{0}:\;x_{n}^{\left(k\right)}=w_{n}^{\left(k\right)},\\ H_{1}:\;x_{n}^{\left(k\right)}=s_{n}^{\left(k\right)}+w_{n}^{\left(k\right)},\\ \;\;\;\;{\left\{f_{0}^{\left(k\right)}\right\}}_{k}\;‘\mathrm{stable}’,\;\mathrm{RW}-\mathrm{like},\\ H_{2}:\;x_{n}^{\left(k\right)}=s_{n}^{\left(k\right)}+w_{n}^{\left(k\right)},\\ \;\;\;\;{\left\{f_{0}^{\left(k\right)}\right\}}_{k}\;‘\mathrm{stable}’,\;\mathrm{not}\;\mathrm{RW}-\mathrm{like},\\ H_{3}:\;x_{n}^{\left(k\right)}=s_{n}^{\left(k\right)}+w_{n}^{\left(k\right)},\\ \;\;\;\;{\left\{f_{0}^{\left(k\right)}\right\}}_{k}\;\mathrm{unstable},\;\mathrm{not}\;\mathrm{RW}-\mathrm{like}. \end{array}\right. \end{array}$$

In terms of AR(1) models, of the form $\xi_{n}=\rho \xi_{n-1}+\varsigma_{n},$${\left\{\varsigma_{n}\right\}}_{n}$ IID, instability can be considered to mean |ρ|>1 and non-RW-likeness can be considered to mean |ρ|≠1 in a statistically significant sense.

The reasoning behind these hypothesis tests will be expanded on, once the necessary tools are extracted from the literature and developed for our problem’s lens (see the discussion at the end of Section 2.3).

### 2.2. Carrier Frequency Estimation

Given that the parameters (*A*, *f*, ϕ) of the signal model are assumed to be unknown, frequency estimation is necessary to apply the hypothesis test introduced in (Equation 9).

Following [39] the carrier frequency of a linear chirp is taken to be the average value of the lower and upper frequencies of the peak region of the Fourier-transformed signal.

The ‘slow’ nature of the RW in frequency (Equation 6) asserts that the endpoints of the peak region are separated by a negligible distance with respect to the size of the frequency space.

Under the assumption of the ‘slowness’ of the RW, the signal’s peak power is distributed across the Fourier bins in a manner that would resemble a narrow peak in Fourier space.

The coarse search process given by Rife and Boorstyn [10] estimates the constant-valued frequency of a single-toned sinusoid. Given the sharpness of the peak in Fourier space, the methods in [10] are an appropriate means of estimating the carrier frequency of the signal.

We note that the estimation process here is not necessarily optimal, but rather is implemented for its well-understood statistical properties.

Taking the $\frac{1}{N}$-normalised Discrete Fourier Transform (DFT)
(10)$$\begin{array}{cc} X_{m}^{\left(k\right)} & =F\left[{\left\{x_{n}^{\left(k\right)}\right\}}_{n}\right]\left(m\right)=\frac{1}{N}\sum_{n=0}^{N-1}x_{n}^{\left(k\right)}\exp\left[\frac{-2\pi inm}{N}\right], \end{array}$$
where m=0,…,N−1 is the Fourier bin number, we estimate the location of the carrier frequency
(11)$$\begin{array}{cc} f_{c}^{\left(k\right)} & =\frac{f_{0}^{\left(k\right)}+f_{0}^{\left(k+1\right)}}{2}, \end{array}$$
by the method in [10], as:(12)$$\begin{array}{cc} {\hat{m}}_{c}^{\left(k\right)} & =\underset{0\leq m\leq N-1}{\mathrm{argmax}}{\left|X_{m}^{\left(k\right)}\right|}^{2}. \end{array}$$

We take ${\hat{f}}_{c}^{\left(k\right)}:=f\left({\hat{m}}_{c}^{\left(k\right)}\right)$, where carets indicate an estimated quantity. We represent the carrier frequency estimator ${\hat{f}}_{c}^{\left(k\right)}$, at fixed *k*, by the ‘true’ carrier frequency of the signal displaced by an estimation error:(13)$$\begin{array}{cc} {\hat{f}}_{c}^{\left(k\right)} & =f_{c}^{\left(k\right)}+\varepsilon^{\left(k\right)}. \end{array}$$

The variance of the peak location of a sinusoidal signal in noise, as estimated according to (Equation 12) and (Equation 13) is shown to be $\sigma_{\varepsilon }^{2}$ = $\frac{1}{N}\frac{6}{{\left(2\pi NT\right)}^{2}\mathrm{SNR}}$ by [40,41] with SNR = $\frac{A^{2}}{\sigma_{w}^{2}}.$

Given limiting distributions of periodogram frequency estimators [42] are normal to leading order, and the asymptotic normality of the Maximum Likelihood Estimator (MLE), we will assume that $\varepsilon^{\left(k\right)}$ follows a distribution of the form $\varepsilon^{\left(k\right)}$ $\overset{a}{∼}$ N(0, $\sigma_{\varepsilon }^{2}$) for sufficiently large *K*. Note, however, that the rate of convergence here is slow—with Theorem 8 of [42] stating effectively that for the standardised version $\bar{\varepsilon }$ of 2πε:(14)$$\begin{array}{cc} P(\bar{\varepsilon }<\xi ) & =\Phi \left(\xi \right)\left[1+\frac{1}{K^{2/5}}\frac{H_{e_{3}}\left(\xi \right)}{3}\right.\\ \; & \;+\frac{1}{K^{4/5}}\left(\frac{H_{e_{4}}\left(\xi \right)}{4}+\frac{H_{e_{6}}\left(\xi \right)}{18}\right)\\ \; & \;+\frac{1}{K^{6/5}}\left(\frac{H_{e_{5}}\left(\xi \right)}{5}+\frac{H_{e_{7}}\left(\xi \right)}{12}+\frac{H_{e_{9}}\left(\xi \right)}{162}\right)\\ \; & \;+\frac{1}{K^{8/5}}\left(\frac{H_{e_{6}}\left(x\right)}{6}+\frac{47H_{e_{8}}\left(x\right)}{480}+\frac{H_{e_{10}}\left(x\right)}{72}\right.\\ \; & \;\left.\left.+\frac{H_{e_{12}}\left(x\right)}{1944}\right)+O\left(K^{-2}\right)\right], \end{array}$$
where $H_{e_{r}}\left(x\right)={\left(-1\right)}^{r}e^{\frac{x^{2}}{2}}\frac{d^{r}}{dx^{r}}e^{-\frac{x^{2}}{2}}$ is the probabilistic Hermite polynomial of order *r*.

As will be noted again later in further discussion, the simulations failed to provide results for SNRs less than −20 dB. We argue that the procedure provides a reasonable degree of performance for Signal-to-Noise Ratios greater than −20 dB, however, in the lower SNR regime, we would argue, that this is likely due to the estimation procedures used as a proof-of-concept for the broader issue of model verification in the problem being considered. More sophisticated estimation approaches could be applied, however, that is not the focus of the paper as this is not an estimation-detection problem we do not need to know whether the model verification lens in this problem can be used, and the SNR-regime of SNRs greater than −20 dB is sufficient to that end.

We do, however, note that Figure A1a,b shows that the parameters (*T*, *K*) do not affect the performance of the procedure as one would anticipate.

### 2.3. Detection of Random Walk Structure

Under $H_{1},$ by (Equation 11) we know that $f_{c}^{\left(k\right)}$ is the average of two RW elements $f_{0}^{\left(k\right)},$$f_{0}^{\left(k+1\right)}$ (*k* fixed) sampled from an RW ${\left\{f_{0}^{\left(k\right)}\right\}}_{k}$ of length *K*. Recalling that the variance of an RW of finite length is its length, then, under $H_{1},$ we assert:(15)$$\begin{array}{cc} V[f_{c}^{\left(k\right)}|H_{1}] & \propto K, \end{array}$$
where we use V[·] to denote the ensemble variance operator.

Now, (Equation 15) tells us that the same simplifications used in the analysis of RWs as in [12] should be used here for ‘regularity’ (in a statistical sense). In line with this, we define the first difference process ${\left\{y_{c}^{\left(k\right)}\right\}}_{k}$ (for k=1,…,K−1),
(16)$$\begin{array}{cc} y_{c}^{\left(k\right)} & :=f_{c}^{\left(k\right)}-f_{c}^{\left(k-1\right)}. \end{array}$$

We also define, at lag $1\leq \tau \leq \frac{K}{2},$ a general difference process ${\left\{\Delta_{\tau }^{\left(k\right)}\right\}}_{k}$ (for k=τ,…,K−τ):(17)$$\begin{array}{cc} \Delta_{\tau }^{\left(k\right)} & :=f_{c}^{\left(k\right)}-f_{c}^{\left(k-\tau \right)}. \end{array}$$

The power of WGN (i.e., $\left\{\right|X_{m}^{\left(k\right)}{{|}^{2}\}}_{m}$ under $H_{0}$) is equally distributed between Fourier bins. Through some algebraic manipulation, we find that, under $H_{0}$:(18)$$\begin{array}{c} {\hat{f}}_{c}^{\left(k\right)}=\left.\left\{\begin{array}{c} -\frac{1}{2T}\\ -\frac{1}{2T}+\frac{1}{NT}\\ \;\;\vdots \\ -\frac{1}{NT}\\ \;\;0\\ \;\frac{1}{NT}\\ \;\;\vdots \\ \;\frac{1}{2T}-\frac{2}{NT}\\ \;\frac{1}{2T}-\frac{1}{NT} \end{array}\right.\right\}\mathrm{with}\;\mathrm{probability}\;\frac{1}{N}. \end{array}$$

Then, under $H_{0},$ by standard definitions and properties of the distributions (see Appendix A), we find that $V\left[{\hat{f}}_{c}^{\left(k\right)}|H_{0}\right]=\frac{1}{12}\left[{\left(\frac{1}{T}\right)}^{2}-{\left(\frac{1}{NT}\right)}^{2}\right].$ And so, by (Equation 16) define,
(19)$$\begin{array}{cc} \sigma_{0}^{2} & :=V\left[{\hat{y}}_{c}^{\left(k\right)}|H_{0}\right]=\frac{1}{6}\left[{\left(\frac{1}{T}\right)}^{2}-{\left(\frac{1}{NT}\right)}^{2}\right]. \end{array}$$

Now, combining (Equation 8), (Equation 11), (Equation 13) and (Equation 16), we may represent ${\hat{y}}_{c}^{\left(k\right)}={\hat{f}}_{c}^{\left(k\right)}-{\hat{f}}_{c}^{\left(k-1\right)}$ (*k* fixed) under $H_{1},$$H_{2},$ or $H_{3},$ by: (20)$$\begin{array}{cc} {\hat{y}}_{c}^{\left(k\right)}= & \frac{1}{2}\left[f\left({\hat{m}}_{0}^{\left(k+1\right)}\right)-f\left({\hat{m}}_{0}^{\left(k-1\right)}\right)\right]\\ \; & +\frac{1}{2}\left[\delta_{0}^{\left(k+1\right)}-\delta_{0}^{\left(k-1\right)}\right]\\ \; & \;\;+\left[\varepsilon^{\left(k\right)}-\varepsilon^{\left(k-1\right)}\right], \end{array}$$

We assume that, under $H_{1},$ we may use a representation of the form:(21)$$\begin{array}{cc} m_{0}^{\left(k\right)} & =m_{0}^{\left(k-1\right)}+u^{\left(k\right)}, \end{array}$$
where the discretised driving term of the RW-like structure is defined to be a sequence of integer-valued jumps between Fourier bins, represented by ${\left\{u^{\left(k\right)}\right\}}_{k}$.

To illustrate the range of behaviours, we consider, at extreme ends, two different forms of ${\left\{u^{\left(k\right)}\right\}}_{k}$. The first case is:(22)$$\begin{array}{cc} u^{\left(k\right)} & =\left.\left\{\begin{array}{c} +1\\ +0\\ -1 \end{array}\right.\right\}\;\mathrm{with}\;\mathrm{probability}\;\approx \frac{1}{3}. \end{array}$$

This discrete uniform jump structure represents the case where minimal information is assumed for the jumps satisfying (Equation 6).

The second case considered is:(23)$$\begin{array}{cc} u^{\left(k\right)} & =\mathrm{nint}\left[z^{\left(k\right)}\right], \end{array}$$
where nint[·] is the nearest integer to [·], and $z^{\left(k\right)}$ ∼ N(0,1).

Then,
(24)$$\begin{array}{cc} \sigma_{1}^{2} & :=V\left[{\hat{y}}_{c}^{\left(k\right)}|H_{1}\right]\\ \; & =\frac{1}{N}\frac{12}{{\left(2\pi NT\right)}^{2}\mathrm{SNR}}+\left\{\begin{array}{c} \frac{9}{24}{\left(\frac{1}{NT}\right)}^{2},\;\;\mathrm{for}\;\mathrm{uniform}\;\mathrm{jumps},\\ \frac{14}{24}{\left(\frac{1}{NT}\right)}^{2},\;\;\mathrm{for}\;\mathrm{normal}\;\mathrm{jumps}. \end{array}\right. \end{array}$$

Thus, for this problem, the ‘slowly’ varying nature, in the sense of condition (Equation 6), of the RW in frequency space under $H_{1},$ is manifest by $\sigma_{0}^{2}>>\sigma_{1}^{2}.$

Given the presence of the discretisation displacement terms in representation (Equation 20), the data is non-gaussian regardless of the jump structure considered. Motivated by the asymptotic properties of variance estimators for non-Gaussian data, as shown in [43], we construct the test statistic,
(25)$$\begin{array}{c} {\hat{\chi }}_{0}^{2}:=\hat{\mathrm{DoF}}\frac{{\hat{\sigma }}^{2}}{\sigma_{0}^{2}}\overset{a}{∼}\chi^{2}\left(\hat{\mathrm{DoF}}\right), \end{array}$$
where $\hat{\mathrm{DoF}}=\frac{2(K-1)}{\hat{\kappa }-\frac{K-4}{K-2}}$ is the sample Degrees of Freedom estimator, with $\hat{\kappa },$${\hat{\sigma }}^{2}$ the sample kurtosis and sample variance estimators respectively, for ${\left\{y_{c}^{\left(k\right)}\right\}}_{k}$. As explained in [44], we require K≳11 for the asymptotic properties in (Equation 25) to hold to at least 1 sigma.

To normalise the test statistic, we perform a Wilson-Hilferty transform,
(26)$$\begin{array}{c} {\hat{Z}}_{0}:=\frac{\sqrt[{3}]{\frac{{\hat{\sigma }}^{2}}{\sigma_{0}^{2}}}-\left(1-\frac{2}{9\hat{\mathrm{DoF}}}\right)}{\sqrt{\frac{2}{9\hat{\mathrm{DoF}}}}}\overset{a}{∼}N\left(0,1\right). \end{array}$$

We reject $H_{0}$ to Probability of False Alarm (PFA) α∈(0,1) if we have that ${\hat{Z}}_{0}<\Phi^{-1}$(α), where Φ(·) denotes the Cumulative Distribution Function (CDF) of the Standard Normal distribution.

However, given that the above procedure seeks *controlled* (in the sense of a ‘slow’ wander as defined in (Equation 6)), but RW-like, jumps in the frequency estimators ${\left\{{\hat{f}}_{c}^{\left(k\right)}\right\}}_{k}$; any stable (in the sense of a mean-reversion type behaviour; |ρ|≤1) random process bounded on a sufficiently small domain would also be flagged by this test as statistically significant. Thus, not only does ${\hat{Z}}_{0}$ distinguish between $H_{0}$ and $H_{1}$, but also $H_{2}$ and $H_{3}$.

The hypothesis problem (Equation 9) addressed here prompts three questions to be tested:Does the measured frequency exhibit a structure resembling an RW?Assuming that the measured frequency does exhibit an RW-like structure, does that structure wander in a ‘slowly’ (in the sense of definition (Equation 6)), rather than randomly jumping in a manner that could be characterised by randomly selecting Fourier bins in a uniformly-distributed manner? That is to say, assuming that the answer to Question 1 was affirmative, then, is that a true RW-like structure, or an artefact of the estimation procedure acting on pure noise?Does the measured frequency revert to the mean, as in an Ornstein-Uhlenbeck process? That is to say, does ${\left\{f_{0}\right\}}^{\left(k\right)}$ wind down? (For good resources on processes like Ornstein-Uhlenbeck processes, we refer the reader to [45]). Mean reversion is likely in astrophysical and GW applications, e.g., signals from rotating neutron stars [46].

In the economics literature, VR-based RWH testing of time series data (such as the LOMAC Test [13]; or the Chow-Denning Test [14]), determines whether the ratio of the variance of the first difference process (Equation 16) to the variance of the difference process (Equation 17) at lag $2\leq \tau \leq \lfloor \frac{K}{2}\rfloor$ (τ fixed) departs from unity in a statistically significant sense. This determines whether mean-reversion behaviours are present, as seen in an Ornstein-Uhlenbeck process.

Whereas alternative RWH testing approaches seek the presence of a unit root against stationarity (such as the DF test [15], or Bhargava’s von Neumann ratio test [47] for the finite sample regime), there are stationary processes that appear like RWs from the lens of unit root testing [48].

It has been shown in [49] (and other papers) that the DF unit root test statistic $\hat{\rho }$ has low power against edge cases (near the unit root) and against trend-stationary processes; resulting in incorrect conclusions being drawn on testing hypotheses for discriminating between an RW and mean-reverting processes (such as Ornstein-Uhlenbeck processes) via unit root tests.

Furthermore, many traditional detectors (such as F test statistics, Hausman test statistics, or Durbin-Watson test statistics) which may be framed to test for (or test against) stationarity tend to yield inconclusive results when tested in an RWH framework [36].

Lastly, it has been shown that the LOMAC and Chow-Denning test statistics are considerably more powerful than the Dickey-Fuller test statistic and other unit root tests [13,14] against even borderline cases such as distinguishing between AR(p) and ARIMA(p,d,q) models; and the Chow-Denning test statistic more powerful than the LOMAC [14].

Thus, for the purposes of this study, we consider VR-based approaches (such as LOMAC or Chow-Denning) and not unit root-based approaches (such as DF).

The first question is one inherent in the RWH testing literature, and so, such methodologies try to address this question.

The second question can be covered by the Controlled Variation Test defined above (Equation 26)—which checks that the first difference process in frequency is varying in a sufficiently controlled manner—that it is more than likely not just randomly picking tones in a uniform manner across the bins.

The third question, similarly to the first question, can be covered by an RWH test in determining if the variance of the lagged differences changes over time in a statistically significant sense.

If the first difference process (Equation 16) is serially uncorrelated, the frequencies do not form an Ornstein-Uhlenbeck process. Then, one could select a sequence $\left\{\tau_{1},\ldots ,\tau_{J}\right\}$ of *J* of lags ($2\leq \tau_{j}\leq \left\lfloor \frac{K}{2}\right\rfloor$), and construct a Chow-Denning [14] test statistic:(27)$$\begin{array}{c} V_{1}=\max_{1\leq j\leq J}\left|M_{1}\left(\tau_{j}\right)\right|, \end{array}$$
where $M_{1}\left(\tau \right)$ is the LOMAC [13] test statistic (2 ≤τ≤$\left\lfloor \frac{K}{2}\right\rfloor$):(28)$$\begin{array}{c} M_{1}\left(\tau \right)=\frac{VR(\tau )-1}{\varphi (\tau )} \end{array}$$
with asymptotic variance φ(τ) defined, as in [12,13] by:(29)$$\begin{array}{c} \varphi \left(\tau \right)=\frac{2(2\tau -1)(\tau -1)}{3\tau }, \end{array}$$
and VR(τ) is the variance ratio defined, as in *loc. cit.* by:(30)$$\begin{array}{c} VR\left(\tau \right)=\frac{\sum_{k=\tau }^{K-\tau }{\left(y_{c}^{\left(k\right)}+\cdots +y_{c}^{\left(k-\tau +1\right)}-\tau \hat{\mu }\right)}^{2}/\tau }{\sum_{k=1}^{K-1}{\left(y_{c}^{\left(k\right)}-\hat{\mu }\right)}^{2}}, \end{array}$$
where $\hat{\mu }$ is the sample mean estimate for ${\left\{y_{c}^{\left(k\right)}\right\}}_{k}$.

One could infer between the two tests, taking similar assumptions of independence as taken by Chow and Denning [14], with appropriate Šidák corrections. Then, to PFA α∈(0,1), we argue that:The tones vary in a sufficiently controlled manner by test (Equation 26) if ${\hat{Z}}_{0}<\Phi^{-1}\left(\alpha^{*}\right)$; and,We reject the RWH by test (Equation 27) if $V_{1}>\Phi^{-1}\left(1-\frac{\alpha^{*}}{2}\right)$,

where $\alpha^{*}=1-{\left(1-\alpha \right)}^{1/J}$ is the Šidák correction for the joint inference of a Chow-Denning Test (Equation 27) of size *J*. *J* is taken as the Šidák correction factor, and not J+1 (as would be implied by the additional independent test from the Controlled Variations test), due to the non-nested nature of the tests [50].

Set hypothesis tests for the Controlled Variations Test (Equation 26) and the (size *J*) Chow-Denning Test (Equation 27), respectively, as: (31)$$\begin{array}{cc} \; & \left\{\begin{array}{c} H_{0}^{1}:\;{\left\{f_{0}^{\left(k\right)}\right\}}_{k}\;‘\mathrm{unstable}’,\\ H_{1}^{1}:\;{\left\{f_{0}^{\left(k\right)}\right\}}_{k}\;‘\mathrm{stable}’, \end{array}\right. \end{array}$$(32)$$\begin{array}{cc} \; & \left\{\begin{array}{c} H_{0}^{2}:\;{\left\{u^{\left(k\right)}\right\}}_{k}\;\mathrm{is}\;\mathrm{serially}\;\mathrm{uncorrelated},\\ H_{1}^{2}:\;\exists \;\mathrm{lag}\;\tau \;\mathrm{so}\;\mathrm{that}\;{\left\{f_{c}^{\left(k\right)}-f_{c}^{\left(k-\tau \right)}\right\}}_{k}\\ \;\;\;\;\;\mathrm{is}\;\mathrm{serially}\;\mathrm{correlated}\;\mathrm{at}\;\mathrm{lag}\;\tau . \end{array}\right. \end{array}$$

The joint inference of the above hypothesis tests determines which of the four hypotheses holds as described in (Equation 9).

Given that the two tests are independent of each other (as per the assumptions that the Chow-Denning test statistic is constructed upon [14]), our testing structure falls under the non-nested hypothesis testing methodologies [50].

The four possible results could arise from this joint inference problem:If the test (Equation 27) asserts that one cannot reject the RWH and the test (Equation 26) asserts that the tones vary in a sufficiently controlled sense: then one could argue that there is an RW in the frequency.If the test (Equation 27) asserts that the data rejects the RWH and the test (Equation 26) asserts that the tones vary in a sufficiently controlled sense: then one could argue that the frequency follows a stable process that does not resemble an RW (an Ornstein-Uhlenbeck process, for example).If the test (Equation 27) asserts that one cannot reject the RWH and the test (Equation 26) asserts that the tones do not vary in a controlled sense: then one could argue there is just noise.If the test (Equation 27) asserts that the data rejects the RWH and the test (Equation 26) asserts that the tones do not vary in a controlled sense: then one could argue that the frequency follows an unstable process that does not resemble an RW.

To elaborate, should the Chow-Denning tests (Equation 27) argue that it is not statistically significant to be able to reject the RWH, then by Definition 2 the central instantaneous frequency process ${\left\{f_{c}^{\left(k\right)}\right\}}^{k}$ as estimated from the observation process ${\left\{x_{n}^{\left(k\right)}\right\}}_{n}^{k}$ is RW-like, that is that it has some component behaving in a manner that is indistinguishable from that of an RW under RWH testing methods.

As RW-like is defined (Definition 2 in a binary sense, as is ‘stability’ (Definition 1), we may take a Fischer-esque philosophical interpretation of the results (i.e., if it is not statistically significant to say that it is A, then it is statistically significant to say it is not A).

The joint inference by the two above tests, are summarised with respect to the hypotheses (Equation 9), (Equation 31) and (Equation 32) in Table 1.

Each case explicitly covers one of the above four possible cases on the constructed problem. Thus, the hypothesis tests as described in (Equation 9) are fully described by the possible interpretations proposed above. It then follows that the above-proposed tests are sufficient to fully specify the considered problem given the assumptions upon which it was constructed.

## 3. Results

### 3.1. Performance of RW Structure Detection

**Proposition** **1.**
*When assuming a normally-distributed jump structure (Equation 23), we propose that, given sufficiently large blockwise sample size N, the controlled variations test, as defined in (Equation 26), asymptotically generates a family of analytical ROC curves ($P_{D}=P_{D}\left(\alpha \right)$, α∈(0,1)):*

(33)
$$\begin{array}{cc} P_{D} & =\Phi \left(\sqrt[{3}]{\frac{\sigma_{0}^{2}}{\sigma_{1}^{2}}}\Phi^{-1}\left(\alpha^{*}\right)+\frac{\left(\sqrt[{3}]{\frac{\sigma_{0}^{2}}{\sigma_{1}^{2}}}--1\right)\left(1--\frac{2}{9(K-2)}\right)}{\sqrt{\frac{2}{9(K-2)}}}\right), \end{array}$$

*on the parameter space (J,N,K,T,SNR)∈$N^{3}$×$R_{>0}^{2}$.*


**Proof.** Under $H_{1},$ the probability distribution $L(y_{c}^{\left(k\right)})$ of ${\left\{y_{c}^{\left(k\right)}\right\}}_{k}$ is a perturbation of the probability distribution $L(u^{\left(k\right)})$ of the process ${\left\{u^{\left(k\right)}\right\}}_{k}$ (scaled by a factor as per (Equation 7)), perturbed by the uniformly-distributed variables ${\left\{\delta_{0}^{\left(k\right)}\right\}}_{k}$. Furthermore, under $H_{1}$, $L({\hat{y}}_{c}^{\left(k\right)})$ is a perturbation of $L(y_{c}^{\left(k\right)})$ by normally-distributed variables ${\left\{\varepsilon^{\left(k\right)}\right\}}_{k}$ [40].Then, by local asymptotic normality [43,51], the Degrees of Freedom estimator $\hat{\mathrm{DoF}}$ should converge to the Gaussian case DoF=K−2, for sufficiently large blockwise sample sizes *N*, if the degree of perturbation is not too large. So, we assert that $\hat{\mathrm{DoF}}\underset{N\to \infty }{⟶}K-2,$ and so, $\frac{\sigma_{0}^{2}}{\sigma_{1}^{2}}{\hat{\chi }}_{0}^{2}\underset{N\to \infty }{⟶}{\hat{\chi }}_{1}^{2}$ where,
(34)$$\begin{array}{c} {\hat{\chi }}_{1}^{2}=\left(K-2\right)\frac{{\hat{\sigma }}^{2}}{\sigma_{1}^{2}}∼\chi^{2}\left(K-2\right). \end{array}$$Two constructions for characterising the jump structure of the RW-like structure on the pivot frequencies ${\left\{f_{0}^{\left(k\right)}\right\}}_{k}$ are considered for the above modelling. We demonstrated the following convergence properties under simulations.As shown in Figure A2a,b, the uniform jump model (Equation 22) for the RW-like structure does not result in a convergence of $\hat{\mathrm{DoF}}\underset{N\to \infty }{⟶}K-2,$ whereas the normal jump model (Equation 23) for the RW-like structure does (see Figure A2c,d).Consider a ‘slow’ branching process generated by normally-distributed variables for the jumps characterising the RW-like structure. Assuming normal jumps (Equation 23), the probability of detection, $P_{D}$, may then be determined. By applying the Wilson-Hilferty transform to (Equation 34), one can relate the test statistic (Equation 26) with respect to the distribution under $H_{1}$ by:
(35)$$\begin{array}{c} {\hat{Z}}_{1}=\sqrt[{3}]{\frac{\sigma_{0}^{2}}{\sigma_{1}^{2}}}{\hat{Z}}_{0}+\frac{\left(\sqrt[{3}]{\frac{\sigma_{0}^{2}}{\sigma_{1}^{2}}}-1\right)\left(1-\frac{2}{9(K-2)}\right)}{\sqrt{\frac{2}{9(K-2)}}}\overset{a}{∼}N\left(0,1\right). \end{array}$$Through algebraic manipulation (see Appendix B) of the test statistic ${\hat{Z}}_{0}$, this generates a family of (asymptotic) analytical ROC curves for the parameters (α, *J*, *N*, *K*, *T*, SNR):
(36)$$\begin{array}{cc} P_{D} & \underset{N\to \infty }{⟶}\Phi \left(\sqrt[{3}]{\frac{\sigma_{0}^{2}}{\sigma_{1}^{2}}}\Phi^{-1}\left(\alpha^{*}\right)+\frac{\left(\sqrt[{3}]{\frac{\sigma_{0}^{2}}{\sigma_{1}^{2}}}-1\right)\left(1-\frac{2}{9(K-2)}\right)}{\sqrt{\frac{2}{9(K-2)}}}\right). \end{array}$$□

Given the analytical performance limits as presented in Proposition 1, we then ask for what sample sizes and signal-to-noise ratios such performance bounds reflect the performance of the model detection methodology considered.

Figure A5a,b reflect the rate of convergence of the performance with respect to increasing blockwise sample sizes, given a signal-to-noise ratio.

As in Figure A5a,b specifically, the analytical performance bounds presented in Proposition 1 represent the performance of the detectors, in practice, in the discrete sample regime (N≤128) for signal-to-noise ratios above −10 dB.

However, simulations failed to provide results for signal-to-noise ratios of −20 dB or less, the underlying reason likely due to non-optimal estimation procedures used (as this was just a proof of concept for the verification method). Further, the simulated ROC curves did not appear to converge to their theoretical counterparts in the very-low SNR regime. Then, it would follow that it is likely that the analytical performance bounds presented in Proposition 1 fail to represent the performance of the detectors for signal-to-noise ratios of −20 dB or less.

Now, the Chow-Denning test statistic $V_{1}$ (Equation 27) of size *J* constructed on the pivot frequencies across a recording of *K* blocks follows a Studentised Maximum Modulus SMM(μ,ν) distribution with parameters μ=J, ν=K−1 [12,14].

Following the same logic that the Chow-Denning test statistic assumes the independence of the LOMAC test statistics from which it was constructed [14], assume the independence of the Chow-Denning test statistic $V_{1}$ (Equation 27) and the Controlled Variations test statistic ${\hat{Z}}_{0}$ (Equation 26). Given this assumption, we may define the detection distribution of the joint tests, under appropriate Šidák corrections to the false alarm rate, as the product distribution $P_{D}^{*}:=P\left({\hat{Z}}_{0}<\gamma_{1},V_{1}>\gamma_{2}|H_{1}\right)$: (37)$$\begin{array}{cc} P_{D}^{*}\left(\alpha \right) & =P\left({\hat{Z}}_{0}<\gamma_{1}|H_{1}\right)P\left(V_{1}>\gamma_{2}|H_{1}\right),\\ \; & =P_{D}\left(\alpha \right)P\left(V_{1}>\Phi^{-1}\left(1-\frac{\alpha^{*}}{2}\right)\right), \end{array}$$
recalling that, as mentioned above, $V_{1}∼SMM\left(J,K-1\right)$.

## 4. Discussion

When performing a model verification test as shown above, to a specified α∈(0,1), the analytical performance bounds (as defined in (Equation 33), and illustrated in Figure A3a and Figure A4) allow one to optimally partition the data across blocks, to better prime the data for analysis, by the following procedure:Determine the sampling rate $f_{s}$ given the Nyquist frequency of the hypothesized signal;Given an SNR >−20 dB, fix *N*∈ [$N_{undersampling}$, $N_{oversampling}$] for which the estimator converges to within a neighborhood less than the discretisation;Take minimal K≥11 (while maximising $J\in \{1,\ldots ,\left\lfloor \frac{K}{2}\right\rfloor -1\}$) such that $P_{D}$ is optimal.

If the signal-to-noise ratio, however, is not greater than −20 dB, then a different estimation procedure would need to be considered.

However, as mentioned in Section 2, the drawback is that the detection method only works given a few key assumptions:The blockwise phases ${\left\{\phi_{n}^{\left(k\right)}\right\}}_{n}$ are quadratic;The model on which the jumps generate the RW-like structure must be standard normal;The period in which the blockwise frequency modulation effectively occurs is known. That is to say, the expected value of ${\left\{t_{0}^{\left(k+1\right)}-t_{0}^{\left(k\right)}\right\}}_{k}$ is known; and,The pivot frequencies ${\left\{f_{0}^{\left(k\right)}\right\}}_{k}$ are uniformly spaced, that var[${\left\{t_{0}^{\left(k+1\right)}-t_{0}^{\left(k\right)}\right\}}_{k}$] is sufficiently small.

The temporal location of (at least) one of the pivots $f_{0}^{\left(k\right)}$ must be known with sufficient accuracy to align the record with the structure in the derivation. That is, the pivot frequencies do indeed occur at ${\left\{t_{0}^{\left(k\right)}\right\}}_{k}$.

However, note that, while the degree of freedom estimator did not converge to the theoretical values under the discrete uniform jump model, the ROC curves still provide intuition on how to partition the data for analysis.

## 5. Conclusions

A model verification test has been designed, which has been shown in simulations to operate as anticipated. The test generalises the temporal RWH testing from the economics literature to test for an RW in frequency, by introducing an additional test to account for phase-wrapping behaviours, and how the peak frequencies of sampled WGN behave.

The use of such a model verification test should reduce the need for more heuristic forms of vetting false-positive detections.

We have yet to explore the impact of biases (such as a mean-reverting behaviour as exhibited under $H_{2}$) on the detection procedure above. Further simulations and modelling are needed to understand how biases affect the ability to determine the presence of RW-like structures in tonal variations.

The above modelling is done for the case where pivots in frequency space are discrete in time. It would be worthwhile to extend the above model verification procedure for Brownian motion-like (BM) structures, with continuous randomness in tonal variations.

Additionally, the above derivations are only formulated for linear blockwise frequency modulations. It would not be complicated to generalise to polynomial blockwise frequency modulations to refine the model verification problem to a SID problem with respect to model order.

Lastly, one normally considers the apparent randomness of behaviours in periodic parameters (such as tones) when interested in forecasting (estimating and/or detecting) the potential of a trend in the signal. That is, for the model presented on the underlying object creating the perceived phenomenon, does the data admit a long-term trend?

A key issue in this problem would be the large number of similar models that could be determined to occur under the same case from the hypothesis structure explored in this paper.

For the following paper, we intend to look into how the question of trend-based behaviours could come into each of the outcomes of the hypothesis test, how one would distinguish between these different trend-based behaviours, and test whether the data support such models.

## Figures and Tables

**Figure 1 sensors-22-06103-f001:**
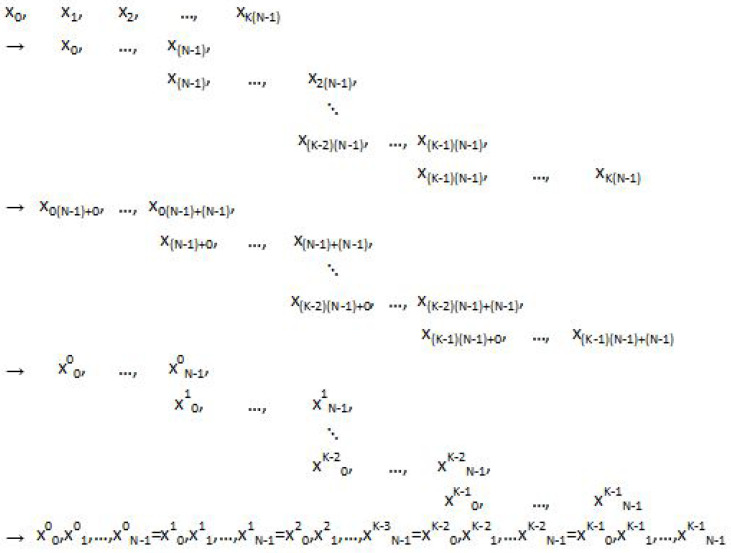
Visualisation of construction of block and blockwise sample indices k,n with k=0,…,K−1,n=0,…,N−1 for partitioning *K* blocks ${\left\{x_{n}^{\left(k\right)}\right\}}_{n}$ (with end-point overlaps $x_{N-1}^{\left(k\right)}=x_{0}^{\left(k+1\right)}$, *k* fixed) of *N* samples from data record ${\left\{x_{n^{\prime }}\right\}}_{n^{\prime }}$ of K(N−1)+1 observations with $n^{\prime }=0,\ldots ,K\left(N-1\right)$.

**Table 1 sensors-22-06103-t001:** Joint inference table.

	$H_{0}^{2}$ Holds	Reject $H_{0}^{2}$
$H_{0}^{1}$ Holds	$H_{0}$ holds	$H_{3}$ holds
Reject $H_{0}^{1}$	$H_{1}$ holds	$H_{2}$ holds

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
