# Peer review of "Testing for a Random Walk Structure in the Frequency Evolution of a Tone in Noise"

_sensors, 2022, doi:10.3390/s22166103_

Round 1

Reviewer 1 Report

Overall, this is an interesting paper. It is written in proper English and pleasant to read. The novelty is properly highlighted. Discussed equations and mathematical formulas seem plausible and free of error. It is visible that the Authors have a good mathematical background. The Appendix provides additional valuable information. The number and quality of cited references is sufficient.

Suggestions and Comments:

There is a square at the bottom of page 5 – why is it so?

The paper undeniably has a research character, Authors perform a lot of statistical analysis, etc. However, do consider presenting some graphical elements, like plots, charts, block diagrams, etc., that would focus the eye of the potential reader.

Consider extending the Conclusions section. Do provide additional feedback and sources of inspiration for other researchers. Do mention about open issues that might inspire others. Point out future directions and upcoming studies.

Consider presenting figs. A1-A22 in larger size, with higher resolution and bigger fonts, so that it is easier to read and interpret. Inserting 4 figures on a single page makes them a bit too small.

To sum up, this paper requires only some minor adjustments before it can be accepted and published.

Author Response

See file attached

Reviewer 2 Report

This paper proposes a test intended for model verification test is developed to determine the presence of a random walk-like structure in the variations in frequency of complex-valued sinusoidal signals measured in additive Gaussian noise.

The paper is well-written and interesting, although I have some remarks,

1.- It seems that this paper fits more in another type of journal, for example in the Mathematics journal, since there it does not deal with any type of sensor.

2.- The novelties and new contributions of the paper are not detailed in the introduction section.

3.- I suggest transfer this paper to Mathematics journal.

Reviewer 3 Report

The authors' proposal is interesting, the paper is well written, and it is relevant in several areas (from structural analysis to gravitational wave). The objective of this work is to test a model to verify the presence of a random-like structure in the variations in the frequency of complex-valued sinusoidal signals in additive noise (Gaussian type).

The authors explain this model mathematically, with a solid formulation. Therefore, the text of the paper itself, as it is a mathematical model, becomes very dense. It does not detract from the merit of the proposal, because it is directly related to signal processing. However, when the figures are only in Appendix C, this becomes a difficult task for readers to follow. For example, figures 1 to 4 can be inserted in Section 2.2 to demonstrate the error behavior for blockwise sample sizes and random walk lengths.

Authors need to be careful with the citation of the figures model, there are parts of the text where it is written Figs or Fig instead of Figure. In addition, the methodology is very explained and detailed.

The weak point is in the discussion because the authors do not make a comparison with similar models present in the literature, nor increase the number of references to problems in which the study can be applied. To have a detailed description, I think that inserting a discussion with similar models and their comparison, in a table format, can facilitate the reading and interest of readers and even highlight the work.

Round 2

Reviewer 2 Report

The authors have replied my questions

Author Response

It seems that we have satisfied, in our revised submission,  all of the issues that this reviewer had with our earlier submission. Many thanks to this reviewer for their important contribution to the quality of our paper.